# Parameter-Efficient Cross-lingual Transfer of Vision and Language Models via Translation-based Alignment

**Zhen Zhang[¶], Jialu Wang[§], Xin Eric Wang[§],**
[¶]UC Santa Barbara, [§]UC Santa Cruz
zhen_zhang@ucsb.edu, {faldict@ucsc.edu, xwang366}@ucsc.edu

## Abstract

Pre-trained vision and language models such as CLIP (Radford et al., 2021) have witnessed remarkable success in connecting images and texts with a primary focus on English texts. Despite recent efforts to extend CLIP to support other languages, disparities in performance among different languages have been observed due to uneven resource availability. Additionally, current cross-lingual transfer methods of those pre-trained models would consume excessive resources for a large number of languages. Therefore, we propose a new parameter-efficient cross-lingual transfer learning framework that utilizes a translation-based alignment method to mitigate multilingual disparities and explores parameter-efficient fine-tuning methods for parameter-efficient cross-lingual transfer. Extensive experiments on XTD (Aggarwal and Kale, 2020) and Multi30K (Elliott et al., 2016) datasets, covering 11 languages under zero-shot, few-shot, and full-dataset learning scenarios, show that our framework significantly reduces the multilingual disparities among languages and improves cross-lingual transfer results, especially in low-resource scenarios, while only keeping and fine-tuning an extremely small number of parameters compared to the full model (e.g., Our framework only requires 0.16% additional parameters of a full-model for each language in the few-shot learning scenario). The codes are available at `https://github.com/eric-ai-lab/PECTVLM`.

## 1 Introduction

Cross-lingual transfer of models facilitates the transfer of learned representations or knowledge from one language to another. It plays a vital role in enhancing the performance of target languages, where the availability of labeled data and linguistic resources are particularly limited. Cross-lingual transfer has found applications in various NLP tasks, including sentiment classification (Chen et al., 2018), dependency parsing (Ahmad et al., 2018), named entity recognition (Rahimi et al., 2019), question answering (Lewis et al., 2019), and dialogue (Schuster et al., 2018), among many others. Recent advancements, such as XLM-R (Conneau et al., 2019), mBART (Liu et al., 2020), and mT5 (Xue et al., 2020), have extended the capabilities of large language models in a multilingual manner, enabling them to comprehend and process multiple languages concurrently.

Two-stream vision-language pre-trained model CLIP (Radford et al., 2021) has demonstrated remarkable performance in image-text retrieval (Cao et al., 2022) by encoding images and text into a shared representation space. However, it primarily focuses on English and cannot comprehend other languages. To address this limitation, Multilingual-CLIP (Carlsson et al., 2022) has been proposed to enhance the CLIP's ability to support multiple languages through cross-lingual transfer. Nevertheless, Multilingual-CLIP treats English as a pivot language, leading to performance disparities across languages, especially low-resource languages. While previous work (Wang et al., 2022) has accessed and highlighted this multilingual disparity, there is currently a lack of proposed solutions to address it.

Multilingual models often encounter a performance trade-off across different languages (Xin et al., 2022) in the sense that overfitting the model in one language may degrade its performance in another. This can be a significant issue as the need to train and maintain separate models for each language can become resource-intensive when dealing with a large number of languages.

The goal of this paper is to address the multilingual disparity in a parameter-efficient manner. To achieve this, we introduce a framework that extends the capabilities of the Multilingual-CLIP model. More specifically, within this framework, we propose a translation-based alignment method

that effectively minimizes the distribution gap between translated and natural language distributions. This alignment method plays a crucial role in significantly reducing the multilingual disparity exhibited by Multilingual-CLIP. Artetxe et al. (2023) also point out the importance of machine translation in classification tasks. Additionally, we adopt Parameter-Efficient Fine-tuning (PEFT) methods (Houlsby et al., 2019; Karimi Mahabadi et al., 2021; He et al., 2022a; Rücklé et al., 2020; Li and Liang, 2021; Guo et al., 2020; Hu et al., 2021; Zaken et al., 2021; Lester et al., 2021a) as a solution to achieve parameter efficiency. Furthermore, we find that, in the zero-shot scenario, hard prompt can also reduce the multilingual disparity and improve multilingual ability in addition to parameter efficiency. Compared with full-model fine-tuning on each language, our framework mitigates the multilingual disparity and obtains higher average performance across all languages, using much fewer additional parameters than a single model.

We conduct our experiments on XTD and Multi30K datasets covering 11 languages in zero-shot, few-shot, and full-dataset learning scenarios. Through extensive analytical experiments, we verify the effectiveness of our framework and provide answers to our research questions. Based on the results of our experiments, we conclude the following:

1. The Multilingual-CLIP model can achieve better performance than the original CLIP model, but still suffers from a significant multilingual disparity. Meanwhile, we find machine translation can map the distribution of text embedding to a better initialization and reduce multilingual disparity. (Section 5.2)
2. Mapping the distribution of text embedding to a better initialization and approximating it to natural pivot language distribution as a better target can significantly help reduce the multilingual disparity. (Section 5.3)
3. PEFT methods can address the excessive resource consumption of Multilingual-CLIP while maintaining acceptable performance degradation. Moreover, we find that hard prompt in English is very effective in the zero-shot learning scenario and can be applied to all languages. (Section 5.4)

## 2 Background

**Multilingual-CLIP** CLIP (Radford et al., 2021), proposed by OpenAI, is a two-stream vision-language pre-trained model with textual and visual encoder. It is trained on a large scale image-text pair dataset using a contrastive loss to encode the image and text into a shared embedding space. CLIP calculate the cosine similarity between image and text features to measure their semantic similarity.

Recently, Multilingual-CLIP (Carlsson et al., 2022) extend CLIP to a multilingual version. This work replaces original English text encoder with a pre-trained multilingual language model such as M-BERT (Devlin et al., 2018) and trains it using teacher learning (Hinton et al., 2015). Although Multilingual-CLIP endowed CLIP with multilingual capabilities, the performance of Multilingual-CLIP in other languages is worse than in English due to the limited amount of data available in low-resource languages, leading to insufficient training in these languages. Furthermore, training data for other languages are translated from English text, which can result in a distribution gap during training and practical application. Noticing this problem, we aim to reduce this multilingual disparity in this paper.

**Parameter-Efficient Fine-tuning** As the size of foundation models (Bommasani et al., 2021) increases, fine-tuning and saving the entire model becomes very resource-intensive. Many parameter-efficient fine-tuning (PEFT) methods have been proposed to solve this issue. These approaches add additional parameters inside the model (Houlsby et al., 2019; Karimi Mahabadi et al., 2021; He et al., 2022a; Rücklé et al., 2020; Li and Liang, 2021), optimize a small portion of the parameters or their low-rank decomposition matrix(Guo et al., 2020; Hu et al., 2021; Zaken et al., 2021), or add trainable token embedding into the input (Lester et al., 2021a). Moreover, He et al. (2021) and Ding et al. (2022) analyze and combine these approaches from a unified perspective. Furthermore, Hu et al. (2022) and Zhang et al. (2022) propose automatic methods to search for an optimal combination of these PEFT methods for language models and visual models, respectively. Many works (Gao et al., 2021; Zhou et al., 2022; Zhang et al., 2021; He et al., 2022b) also apply PEFT methods to CLIP models. Nevertheless, those PEFT methods have not been thor-

oughly explored for the Multilingual-CLIP model in the cross-lingual transfer setting. It is important to note that PEFT methods often result in a decline in performance to varying degrees compared to full-model fine-tuning. Therefore, it is essential to conduct experiments to verify the effectiveness of these methods and determine the most appropriate approach for specific tasks and models.

# 3 Framework

Our main contribution is a cross-lingual transfer framework for Multilingual-CLIP (Figure 1). In this framework, we propose a novel translation-based cross-lingual alignment method that reduces the multilingual disparity and exploits parameter-efficient tuning methods to solve the resource consumption problem in cross-lingual transfer.

## 3.1 Insights of Our Method

Our method is grounded in experimental results (Table 2) and motivated by the work of (Wang et al., 2022). Table 2 shows that machine translation improves M-CLIP's multilingual disparity and performance with English consistently performing the best. Wang et al. (2022) demonstrates that text embeddings are more closely aligned in the translation portion. Our hypothesis is that English text can yield embeddings of better quality owing to rich language resource, thus become a good target for alignment. Additionally, translation can bring other languages' embedding to a better initialization for further alignment with English.

## 3.2 Translation-based Cross-lingual Alignment

In Figure 1(a), we present a diagram of the translation-based cross-lingual alignment method. The blue circles represent the distribution of natural language embeddings in the representation space, while the orange ones represents the embedding distribution of text generated by machine translation. Since the pre-training of Multilingual-CLIP involves aligning English text with translated target text in other languages, there exists a disparity in the distribution of text between training and real-world usage. This gap varies among different languages and contributes to the multilingual discrepancy. In our approach, we aim to minimize this distribution gap by employing machine translation to map one embedding distribution to another. We propose an alignment method using pivot-target

language text pairs, which depict the same image in both the pivot (English) and target languages.

Our alignment method has different combinations of routines and loss functions to be compared.

**Alignment Routines.** The term "alignment routine" refers to which two distinct embedding types we are going to align. Given the pivot language (English) and target language, there are three routines in the representation space to narrow the gap between embedding distributions as shown in the left part of Figure 1. To be specific, these routines are (1) aligning original English and target language text embeddings, (2) aligning translated English (to target) and original target language text embeddings, as Multilingual-CLIP is only pre-trained in target language in translation distribution where it performs better than natural language distribution, and (3) aligning original English and translated target language (to English) text embeddings. We compare these three routines in the experiments and find routine 3 performs best. Note that these three routines do not apply to pivot languages (English) and routine 3 still need machine translation in the inference process.

**Alignment loss functions.** In addition to the alignment routines, alignment loss functions must also be considered. Mean Squared Error (MSE) loss and contrastive loss are two practical loss functions for narrowing the distance between embeddings.

To be specific, the original contrastive loss between image and text embeddings can be written as:

$$\mathcal{L}_{\text{i2t}} = -\frac{1}{N} \sum_{i=1}^{N} \log \frac{e^{\cos(v_i, t_i)/\tau}}{\sum_{j=1}^{N} e^{\cos(v_i, t_j)/\tau}}, \quad (1)$$

$$\mathcal{L}_{\text{t2i}} = -\frac{1}{N} \sum_{i=1}^{N} \log \frac{e^{\cos(t_i, v_i)/\tau}}{\sum_{j=1}^{N} e^{\cos(t_i, v_j)/\tau}}, \quad (2)$$

where $v_i$ is the visual embedding of the image in the $i$-th pair and $t_j$ represents the textual embedding of the text in the $j$-th pair. We use i2t and t2i to represent image-text and text-image matching. $\tau$ is the temperature used to scale the cosine similarity. Following CLIP (Radford et al., 2021), it is set to 0.01. N is the number of image-text pairs in the dataset. Pivot and target language text embedding can be obtained through text encoder:

$$T^{\text{pivot}} = \text{text\_encoder}(\text{text}^{\text{pivot}}), \quad (3)$$
$$T^{\text{tgt}} = \text{text\_encoder}(\text{text}^{\text{tgt}}), \quad (4)$$

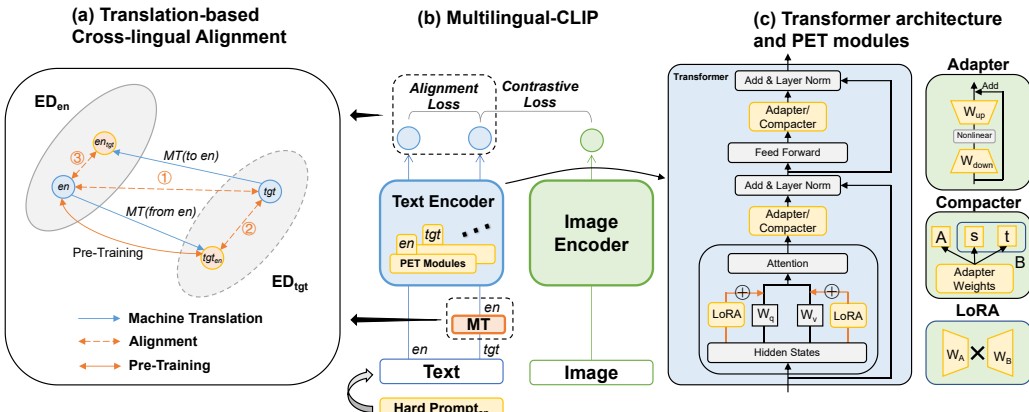

Figure 1: Illustration on our framework based on multilingual CLIP. We propose a translation-based alignment method to narrow the distribution gap and adopt PEFT methods to achieve parameter efficiency. We also find hard prompt is very effective in the zero-shot scenario.

Note that texts can be translated from one to another:

$$\text{text}^{\text{tgt}} \leftarrow \text{Trans}_{\text{tgt} \rightarrow \text{en}}(\text{text}^{\text{tgt}}), \quad (5)$$

$$\text{text}^{\text{pivot}} \leftarrow \text{Trans}_{\text{en} \rightarrow \text{tgt}}(\text{text}^{\text{pivot}}). \quad (6)$$

These two different losses to regularize the distance between parallel text embeddings can be represented as:

$$\mathcal{L}_{\text{alignment}}^{\text{MSE}} = \frac{1}{N} \sum_{i=1}^{N} (t_i^{pivot} - t_i^{target})^2, \quad (7)$$

$$\mathcal{L}_{\text{alignment}}^{\text{contrastive}} = -\frac{1}{N} \sum_{i=1}^{N} \log \frac{e^{\cos(t_i^{\text{pivot}}, t_i^{\text{tgt}})/\tau}}{\sum_{j=1}^{N} e^{\cos(t_i^{\text{pivot}}, t_j^{\text{tgt}})/\tau}}$$
$$- \frac{1}{N} \sum_{i=1}^{N} \log \frac{e^{\cos(t_i^{\text{pivot}}, t_i^{\text{tgt}})/\tau}}{\sum_{j=1}^{N} e^{\cos(t_j^{\text{pivot}}, t_i^{\text{tgt}})/\tau}}. \quad (8)$$

$t_i^{pivot}$ and $t_i^{target}$ refers to the i-th text embedding in pivot (English) and target language. They are added to the contrastive loss between image and text with an alignment coefficient $\lambda$:

$$\mathcal{L} = \mathcal{L}_{\text{i2t}} + \mathcal{L}_{\text{t2i}} + \lambda \cdot \mathcal{L}_{\text{alignment}}. \quad (9)$$

### 3.3 Parameter-Efficient Cross-lingual Transfer Learning

Our framework utilizes Parameter-Efficient Fine-tuning (PEFT) methods to solve the resource consumption problem. We compare the following PEFT methods with full-model fine-tuning. When training these PEFT modules, we freeze the parameter of Multilingual-CLIP. PEFT methods are usually designed for different tasks, while we instead use them for different languages to achieve parameter efficiency.

**Adapter (Houlsby et al., 2019):** Figure 1(c) top right. Adapter adds a few trainable linear neural modules after every attention and feedforward layer. it consists of a down sampling matrix $W_{down} \in \mathbb{R}^{d \times r}$ and a up sampling matrix $W_{up} \in \mathbb{R}^{r \times d}$ with a nonlinear activation function $f$ in the middle, where $d$ is the dimention of the input $x \in \mathbb{R}^d$ of the adapter. There is also a residual connection and the output $O$ can be written as:

$$O = x + f(xW_{down})W_{up} \quad (10)$$

**Compacter (Karimi Mahabadi et al., 2021):** Figure 1(c) center right. An improvement work of Adapter. This work replaces the standard Adapter layer with a low-rank hypercomplex Adapter layer, which requires fewer parameters and yields competitive results. To be specific, Compacter decompose the $W_{down} \in \mathbb{R}^{d \times r}$ to the sum of $k$ Kronecker products of matrix $A_i \in \mathbb{R}^{k \times k}$ and $B_i \in \mathbb{R}^{\frac{d}{k} \times \frac{r}{k}}$. $B_i$ is further decomposed to two low-rank matrices $s_i \in \mathbb{R}^{\frac{d}{k} \times r_B}$ and $t_i \in \mathbb{R}^{r_B \times \frac{d}{k}}$, where $r_B$ represents the rank of $B_i$. Finally, the formula can be written as follows:

$$W_{down} = \sum_{i}^{k} A_i \otimes B_i = \sum_{i}^{k} A_i \otimes (s_i t_i) \quad (11)$$

$W_{up}$ is also decomposed in this way with shared $A_i$.

**LoRA (Hu et al., 2021):** Figure 1(c) bottom right. LoRA assumes the low-rank change of model weights $W \in \mathbb{R}^{d \times K}$ and then uses two trainable

rank-decomposition matrices $W_A \in \mathbb{R}^{d \times r}$ and $W_B \in \mathbb{R}^{r \times k}$ to approximate the matrix change. Consequently, LoRA adds changes to the original output $O$ from the input $x$:

$$O \leftarrow O + x W_A W_B \qquad (12)$$

Following the default setting of LoRA, we apply this method to query and value projection matrices $(W_q, W_v)$ in self-attention layers.

**Hard and soft Prompt (Lester et al., 2021b):** Figure 1(b) bottom. Hard prompt attaches text prompts to the front of the input text (e.g., "a photo of [Text]"), which is manually designed and explainable. CLIP uses multiple hard in the pretraining phase, so we are interested in whether it is applicable in cross-language transfer scenarios. We compare different combinations of the hard prompt and input text in different languages, and find English hard prompt work well on average across all languages. Soft prompt, also known as prompt tuning, adds trainable token embeddings to the front of the input. We do not plot soft prompt in our framework as our experiments show it doesn't perform well.

In our experiments, we also tune the linear head and layer-norm layer of the text encoder when we train Adapter, Compacter and LoRA and the number of their parameters is 0.44%, 0.05% and 0.16% of the text encoder, respectively. Hard prompts only require saving a few words, while soft prompts require storing several token embeddings, each of which takes up 1024 floating point numbers in storage space.

### 3.4 Optimization Objective

Our optimization objective is to get optimal parameters of specific PEFT modules on each target language by minimizing the loss $\mathcal{L}$.

## 4 Experimental Setup

**Dataset** We work with two datasets: (1) Flickr30K (Young et al., 2014) is an image captioning datasets in English, split into train/dev/test datasets with the number of 29000/1024/1000. The Multi30K (Elliott et al., 2016) dataset extends captions of Flickr30K dataset with human translated and independent German sentences. Elliott et al. (2017) and Barrault et al. (2018) further translate English Flickr30k captions to French and Czech, respectively. (2) XTD (Aggarwal and Kale, 2020)

| Framework(PPL) | XTD | | | | Multi30K | | | |
|---|---|---|---|---|---|---|---|---|
| | en↑ | Avg.-en↑ | Std↓ | Range↓ | en↑ | Avg.-en↑ | Std↓ | Range↓ |
| | Zero-shot | | | | | | | |
| w/o (M-CLIP,100%) | 63.44 | 57.42 | 4.75 | 16.00 | 66.65 | 65.03 | 0.93 | 2.15 |
| w/ (Ours, <0.45%) | **64.06** | **59.94** | **3.26** | **10.84** | **67.80** | **66.73** | **0.59** | **1.30** |
| | Few-shot | | | | Full-dataset | | | |
| w/o (M-CLIP,100%) | 64.67 | 58.57 | 4.83 | 16.06 | **76.10** | 74.93 | 0.70 | 1.55 |
| w/ (Ours, <0.45%)) | **64.83** | **60.45** | **3.10** | **10.61** | 75.35 | **75.65** | **0.56** | **1.25** |

Table 1: Results on XTD and Multi30K of Multilingual-CLIP (M-CLIP) with and without our framework. We report the Recall@1 score and bold the best result in each scenario on each dataset. "Avg.-en" represents the average score without English and "PPL" represents parameters per language. Statistical indicators standard deviation (Std) and range are used to evaluate multilingual disparity. PEFT methods used for zero-shot, few-shot and full-dataset scenarios are hard prompt, LoRA and Adapter, respectively.

is a Cross-lingual dataset for the image-text retrieval task covering 11 languages. It only has a test split with 1000 samples per language with the same images. For few-shot setting, we randomly split the original test split into train/dev/test sets with 50/50/900 image-text pairs.

**Base Model and Translation Tool** We use XLM-$R_{Large}$-ViT$_{L/14}$ (Carlsson et al., 2022) as our base model. The model fixes the original visual encoder of OpenAI ViT$_{L/14}$ (Radford et al., 2021) and replaces the text encoder by XLM-Roberta$_{Large}$ (Conneau et al., 2019) trained by teacher learning (Hinton et al., 2015). We use Google Translation[1], a strong Neural Machine Translation (NMT) system, to translate between all the different languages. To reduce the computational overhead, we translate the dataset in advance, rather than when it is used. We give the results of the translation in the code part. Analysis of different machine translation tools can be seen in Appendix B.

## 5 Experiments and Analysis

In this section, we first present the overall results of our framework with the optimal combination and then conduct analytical experiments to demonstrate the effectiveness of machine translation (Section 5.2), routine 3 and MSE loss is the best choice for alignment (Section 5.3) and hard prompt, LoRA and Adapter is respectively outperforms in zero-shot, few-shot and full-dataset scenario (Secion 5.4). The details of our experimental configurations are in Appendix A.

## 5.1 Cross-Lingual Transfer Results on XTD and Multi30K

Table 1 shows the results of Multilingual-CLIP with and without our framework on XTD and Multi30K datasets in zero-shot, few-shot, and full-dataset scenarios. We report the Recall@1 score on the English dataset and the average score across all other languages for evaluation of the multilingual performance and calculate statistical indicators, standard deviation, and range for evaluation of the multilingual disparity.

Compared to the vanilla Multilingual-CLIP, our framework outperforms in all zero-shot, few-shot, and full-dataset scenarios. It reduces range by more than 5 points and standard deviation by more than 1.5 points while achieving significant performance improvement both in English and on average across all other languages on the XTD dataset in both zero-shot and few-shot scenarios, which is a common application scenario of low-resource languages. The improvement in the Multi30K dataset is also significant. In terms of the number of parameters, our framework is also more efficient than full-model fine-tuning. In eleven languages, Adapter and LoRA only use 4.89% and 1.73% of the parameters respectively, which is far less than the parameters of 11 models.

To sum up, our framework significantly reduces the multilingual disparity and enables parameter-efficient cross-lingual transfer, with a byproduct of improving multilingual performance.

## 5.2 Analysis on Multilingual Disparity of Multilingual CLIP

Considering that Multilingual-CLIP replaces the text encoder with a multilingual version while keeping the image encoder of CLIP, we directly compare Multilingual-CLIP to the CLIP model equiped with machine translation as a strong baseline (Jain et al., 2021). We compare the performance of the CLIP and Multilingual CLIP models on XTD dataset with the help of machine translation. we utilize machine translation to convert non-English corpora into English. This translated version is then employed as input for both CLIP and Multilingual-CLIP. Additionally, we perform translations of the English dataset into each respective language and evaluate them using multilingual CLIP.

**Result analysis.** As shown in Table 2, we observe that although the original CLIP has lim-

¹https://translate.google.com/

ited multilingual capabilities, with the help of machine translation, it can achieve high multilingual capabilities to a certain extent and even surpass Multilingual-CLIP. However, Multilingual-CLIP with machine translation obtain the best multilingual capability and the lowest disparity in both scenarios. We did not use the setting "M-CLIP (en→tgt)" as a comparison as it is not a practical application scenario. On the other hand, there is a large multilingual disparity for multilingual CLIP. For example, the difference between Japanese and English is up to 16% and the standard deviation is up to 4.7%. Aided by machine translation, the improvement of Multilingual-CLIP on multilingual differences is very obvious. Finally, using data translated from English, Multilingual-CLIP shows a large improvement in other languages (mean improvement of 1.6%) but still lower than in English. This may be because the model is pre-trained with text translated from English, making it more adapted to this situation. This also suggests that multilingual disparity are partly the result of differences in the quality of the datasets in different language instead of the ability of the model.

**Remark.** In terms of the multilingual representation space, machine translation serves the purpose of mapping text from one embedding distribution to another. By mapping text into English, we achieve a more favorable initialization of the distribution for subsequent alignment processes. While the embedding distribution of the translated text may differ slightly from that of natural language, this disparity is significantly smaller compared to the distinction between two distinct languages. Consequently, it becomes easier to optimize and narrow the gap between these distributions.

## 5.3 Analysis on How to Exploit Pivot Language

Datasets in low-resource languages are usually small, and we can obtain the corresponding pivot language (English) text from target language text through human annotation. In particular, for the image caption dataset, annotators can directly give high-quality English captions based on the image without mastering other languages. For (relatively) high-resource language, the parallel text is also a source to obtain texts in different languages with the same meaning. We call these texts as pivot-target language text pair. Since the model has higher performance on pivot language and those

| Method | | XTD | | | | | | | | | | | | | | |
|---|---|---|---|---|---|---|---|---|---|---|---|---|---|---|---|---|
| | | en | de | fr | es | it | ko | pl | ru | tr | zh | jp | Avg.↑ | Avg.-en ↑ | Std↓ | Range↓ |
| Zero-shot | CLIP | 62.06 | 25.33 | 32.94 | 31.28 | 25.00 | 0.56 | 5.67 | 1.72 | 4.50 | 1.39 | 6.83 | 17.93 | 13.52 | 19.36 | 61.50 |
| | CLIP + MT | 62.06 | 60.56 | 60.56 | 59.72 | 58.78 | 58.00 | 60.11 | 53.72 | 60.06 | 56.72 | 50.72 | 58.27 | 57.90 | 3.38 | 11.34 |
| | M-CLIP | **63.44** | 59.94 | 60.06 | 58.90 | 60.72 | 51.00 | **61.50** | 56.11 | 59.28 | 59.28 | 47.44 | 57.97 | 57.42 | 4.75 | 16.00 |
| | M-CLIP (en→tgt) | **63.44** | **62.59** | **62.39** | **62.61** | 61.33 | 61.33 | 61.33 | **61.78** | 62.11 | **62.44** | 54.17 | **61.41** | **61.21** | **2.49** | **9.27** |
| | M-CLIP + MT | **63.44** | 61.11 | 61.33 | 62.33 | 61.17 | **61.44** | **61.50** | 54.83 | 61.67 | 58.00 | 52.72 | 59.96 | 59.61 | 3.35 | 10.72 |
| Few-shot | M-CLIP | **64.67** | 61.83 | 60.67 | 61.33 | **62.00** | 52.22 | 62.61 | **56.33** | 60.39 | **59.72** | 48.61 | 59.13 | 58.57 | 4.83 | 16.06 |
| | M-CLIP + MT | **64.67** | **61.89** | **62.00** | **62.17** | 61.44 | **61.00** | **63.00** | 56.11 | **62.00** | 59.50 | **53.83** | **60.69** | **60.29** | **3.14** | **10.84** |

Table 2: Recall@1 in percentage on image-text retrieval XTD dataset. We compare CLIP and Multilingual-CLIP with machine translation as a tool in zero-shot and few-shot scenarios. We average the score to get the overall performance across the languages, and evaluate the multilingual disparity with standard deviation and range. M-CLIP is short for Multilingual-CLIP and MT is short for machine translation. We bold the best scores for zero-shot and few-shot respectively. "Avg.-en" represents average score without English and "en→tgt" means data in other languages is translated from English set.

| Setting | XTD | | | Multi30k | | |
|---|---|---|---|---|---|---|
| | Avg.-en ↑ | Std↓ | Range↓ | Avg-en ↑ | Std↓ | Range↓ |
| | Few-shot (en: 64.67) | | | Full (en: 76.10) | | |
| - | 58.57 | 4.83 | 16.06 | 74.93 | 0.70 | 1.55 |
| MT | 60.29 | 3.14 | 10.84 | 75.30 | 0.68 | 1.45 |
| Routine1+MSE | 58.85 | 4.72 | 15.73 | 75.57 | 0.72 | 1.55 |
| Routine2+MSE | 58.81 | 4.75 | 15.67 | 75.65 | 0.95 | 2.10 |
| Routine3+MSE | **60.52** | **3.11** | **10.50** | **75.97** | **0.60** | 1.45 |
| Routine3+CL(pivot-tgt) | 60.30 | 3.12 | 10.73 | 75.47 | 0.98 | 2.15 |
| Routine3+CL(pivot-image) | 60.46 | 3.12 | 10.56 | 75.85 | 0.61 | **1.45** |

Table 3: We compare different alignment routines for parallel corpus on XTD and Multi30K datasets. We report the score in English once as these combinations cannot applies on English set. CL is short for contrastive loss. We bold the best results on each dataset.

pivot-target text pairs can provide more information, there must be a better approach to exploit pivot language for better cross-lingual transfer of Multilingual-CLIP.

**Comparison of different alignment routines and loss functions.** In section 3.2, we mention three alignment routines and two alignment loss functions. We compare their different combinations with two baselines without alignment. In the first baseline, we directly apply contrastive loss between images and texts in target language in a mini-batch. For the second baseline, we translate all texts to English previously on the basis of the first baseline. We conduct experiments on XTD and Multi30K in zero-shot, few-shot, and full-dataset learning scenarios.

As shown in Table 3, we first compare different routines combined with MSE loss and find that routine 3, translating the target language into English and doing alignment between natural and translation English embedding distribution, performs best. Natural refers to "generated by humans rather than machine translation". Then we compare different alignment methods with routine 3 and find MSE loss performs best. Ultimately, routine3 combined with MSE loss performs best on all 3 metrics.

This can be explained as that Multilingual-CLIP uses MSE loss for text-text pairs in the pre-training stage, and natural English embedding distribution is a better distribution, which denotes an embedding with higher performance. Meanwhile, machine translation maps the target language embedding distribution to a distribution close to optimal distribution where multilingual CLIP performs best, making it easier to optimize.

## 5.4 Analysis on Parameter-Efficient Cross-Lingual Transfer Learning

In this section, we first evaluate the performance of hard prompt. Then we compare Adapter, Compacter, and LoRA, and discuss the feasibility of using these methods for cross-lingual transfer.

### 5.4.1 Hard Prompt

We primarily focus on the prompting method, in particular the hard prompt method, which demonstrates remarkable capabilities in zero-shot learning scenarios where fine-tuning model parameters with domain-specific data is unfeasible. In a multilingual setting, we must take into account two types of prompts. Firstly, prompts can be constructed in multiple languages, potentially leading to performance disparities across different languages. Secondly, the text input can be automatically translated into any other language using machine translation. Therefore, we explore the following combinations: (1) Simply prepend an English prompt before the text. (2) Translate the optimal English prompt into target languages and append them before their respective texts. (3) Translate all the text inputs to English and place English prompts in the front.

**Result analysis.** From results in Table 4, it can be found that the zero-shot performance increases by simply adding prompt in English, with both

| Setting | en | de | fr | es | it | ko | pl | ru | tr | zh | jp | Avg.↑ | Avg.·-en ↑ | Std↓ | Range↓ |
|---|---|---|---|---|---|---|---|---|---|---|---|---|---|---|---|
| | | | | | | Zero-shot | | | | | | | | | |
| (1) English Hard Prompt | **64.06** | 60.72 | 59.89 | 61.78 | 61.28 | 51.67 | 62.06 | **56.00** | 60.11 | **59.78** | 47.83 | 58.65 | 58.11 | 4.90 | 16.23 |
| (2) Target Lang Hard Prompt | **64.06** | 60.83 | 60.33 | 62.06 | 61.22 | 49.00 | 61.00 | **56.00** | 59.39 | 59.33 | 48.06 | 58.30 | 57.72 | 5.22 | 16.00 |
| (3) English Hard Prompt + MT | 64.06 | **61.5** | 61.89 | **62.28** | **61.56** | 60.39 | 62.39 | 55.5 | **61.89** | 58.78 | 53.22 | **60.31** | **59.94** | 3.26 | 10.84 |
| | | | | | | Few-shot | | | | | | | | | |
| Soft Prompt (3 tokens) | 63.94 | 61.17 | **62.00** | 62.17 | **61.56** | **60.50** | **62.44** | 55.44 | 61.83 | 58.72 | **53.33** | 60.28 | 59.92 | **3.22** | **10.61** |
| Soft Prompt (20 tokens) | 62.72 | 59.89 | 60.50 | 60.28 | 59.83 | 59.44 | 60.22 | 54.33 | 60.00 | 58.11 | 53.11 | 58.95 | 58.57 | 2.81 | 9.61 |

Table 4: Results on XTD dataset for comparison of different combinations of prompts and texts.

| Setting | Updated Params per Lang (%) | XTD (few-shot) | | | | | | | | | | | | Multi30K (Full-dataset) | | | | |
|---|---|---|---|---|---|---|---|---|---|---|---|---|---|---|---|---|---|---|
| | | en | de | fr | es | it | ko | pl | ru | tr | zh | jp | Avg | en | cs | de | fr | Avg |
| FT | 100 | 64.67 | 61.83 | 60.67 | 61.33 | 62.00 | 52.22 | 62.61 | 56.33 | 60.39 | 59.72 | 48.61 | 59.13 | 76.10 | 74.55 | 74.80 | 75.45 | 75.23 |
| Adapter | 0.45 | 64.44 | **61.17** | **60.61** | 60.72 | 61.39 | 52.17 | **62.83** | 56.44 | 59.50 | 59.67 | 49.17 | 58.92 | 75.20 | **74.50** | 73.85 | **76.40** | **74.99** |
| Compacter | 0.05 | 63.67 | 60.61 | 60.44 | 60.50 | 61.33 | 52.06 | 62.67 | 55.94 | 59.72 | 59.39 | 49.00 | 58.67 | 74.25 | 72.40 | 73.55 | 73.85 | 73.51 |
| LoRA | 0.16 | **64.83** | 61.10 | 60.39 | **60.89** | **61.67** | **53.06** | 61.94 | **56.67** | 60.50 | 59.82 | 49.56 | 59.13 | 75.35 | 72.30 | 73.65 | 74.85 | 74.04 |
| Our framework | 0.16 / 0.45 | 64.50 | **61.94** | **62.00** | **62.00** | 61.83 | **61.65** | 62.72 | 55.94 | 62.33 | 59.84 | 54.22 | **60.82** | 75.35 | 75.40 | **75.15** | **76.40** | **75.58** |

Table 5: Comparison of different PEFT methods. We care about the degree of performance degradation caused by different PEFT methods. We bold the best score among the three PEFT methods and bold the score of our framework if it is better than full-model fine-tuning. Our framework updates 0.16% and 0.45% parameters for few-shot and full-dataset scenario, respectively. The non-"Our framework" rows show results without translation alignment.

text input in target language and translated into English. We compare different hard prompts and find "a photo of" performs best. However, Translating English prompt into target language makes the performance decrease slightly. Finally, we get the best performance by translating all the text inputs into English and adding the best English prompt to them.

**Comparison with soft prompt.** We also conduct experiments on soft prompt based on the third combination: initiate prompt from the best prompt, which result in 3 trainable token embeddings, and randomly initiate 20 token embeddings. With 50 training instances in the few-shot scenario, the model obtains marginal improvement or even performance decreasing by utilizing these templates. As a result, we do not incorporate soft prompt in our framework.

### 5.4.2 Other PEFT Methods

In this section, we compare three popular PEFT methods, Adapter, Compacter, and LoRA, with full-model fine-tuning. Following He et al. (2022a), we unfreeze the linear head and assign the same learning rate as fine-tuning. The number of parameters of Adapter, Compacter and LoRA is only 0.45%, 0.05% and 0.16% of Multilingual-CLIP's text encode, respectively. Thus, even if we assign different parameters to each of the 100 languages, the total number of parameters is smaller than that of a single model. As a result, we utilized language-specific modules, yet our method can still achieve

parameter efficiency and outperform full-model fine-tuning. From the results shown in Table 5, we can find that Adapter performs the best in full-dataset scenario, which preserves 99.7% performance of fine-tuning and LoRA performs the best in few-shot scenario, which even achieve almost the same performance. We further adopt Adapter and LoRA with the best combination discussed in Section 5.3, which forms a part of our final framework, and find this combination obtains better performance than full-model fine-tuning. It indicates that with the help of PEFT methods, our framework can achieve the parameter-efficiency without losing too much performance.

### 5.4.3 Summary of PEFT Methods

In summary, this section presents three primary contributions of PEFT methods: (1) We conducted a comprehensive investigation of PEFT methods in cross-lingual transfer learning scenarios using the M-CLIP model. Through this study, we discovered novel insights. For instance, in multimodal and cross-lingual transfer settings, we found that adapters are often not the optimal solution, and employing English prompts tends to yield superior performance compared to multilingual prompts. (2) Our framework demonstrated improved parameter efficiency compared to full-model fine-tuning, especially when considering the vast array of available languages. (3) We achieved enhanced transfer learning performance, a significant accomplishment considering that PEFT methods generally exhibit lower performance than full-model fine-

tuning in other research works.

## 6 Conclusion

In this paper, we propose a framework that significantly mitigates the multilingual disparity of Multilingual-CLIP in a parameter-efficient manner in zero-shot, few-shot and full-dataset scenarios. Our framework uses a translation-based alignment method and adopts parameter-efficient tuning methods. Analytical experiments indicate that machine translation is effective for cross-lingual transfer; exploiting pivot language can help reduce the disparity; parameter-efficient tuning methods are beneficial for reducing resource consumption without too much performance degradation.

## 7 Limitations

Our work primarily focuses on addressing multilingual disparity by improving the multilingual text encoder in the CLIP-liked framework. However, it is very possible that the visual encoder can also be enhanced with image and text data from diverse culture and languages. During the pre-training process, the original Multilingual-CLIP's visual encoder is directly aligned with English corpora only, and connected with other languages by using English as a pivot language. We expect future work can address the multilingual disparity problem from the perspective of a more powerful visual encoder.

## Broader Impact

This work provides a framework for cross-lingual transfer in few-shot learning setting. The deployment of our method is potential to mitigate the performance disparity for state-of-the-art multimodal models for scarce-resource languages. However, we note that our method relies on the collection of parallel corpus, either collected from online machine translation systems or native human speakers. Our work does not thoroughly scrutinize whether these parallel corpus contains implicit social biases in different dimensions, such as race, gender and religion. When these parallel corpus contains unexpected biases or stereotypes, it is likely that the model learned from such data may perpetuate these biases that we did not foresee.

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

## A   Experimental Details

We use Adam (Kingma and Ba, 2014) as our optimizer with a cosine decay learning rate scheduler. For few-shot on XTD dataset, we train the model for 40 epochs and use the batch size of 10, which results in a total of 200 steps. When we train our model on the whole Multi30K, we set the number of epoches to 15 with batch size of 48 due to the memory limitation and the total steps is 9k. When the memory is insufficient, we appropriately reduce batchsize to adapt. We evaluate the performance every 5/300 steps respectively and save the best checkpoint for test. For a more obvious comparison, We report Recall@1 score in all experiments. We froze the image encoder for a fair comparison since the difference between languages is in the text input and tuning image encoder will introduce additional randomness. All the experiments are done on 8 Nvidia V100 GPUs. We use the official code of CLIP and Multilingual-CLIP to load and use pre-training parameters

To find an optimal combination of hyperparameters, we conduct a grid search on learning rate and guidance coefficient $\lambda$. The learning rates lie within {3e-5, 1e-4, 3e-4} for parameter-efficient methods, and {1e-6, 3e-6, 1e-5} for fine-tuning the whole model. The guidance coefficients fall within a large range from 0.001 to 10. The optimal $\lambda$s vary from language to language, but most of them are distributed in a smaller interval from 0.1 to 1.

## B   Analysis on Different Machine Translation Tools

Different machine translation models may have various impacts on the experimental outcomes. Our decision to use Google Translate was based on its widespread adoption and high-quality translation results, which reflect the general situation. Higher-quality professional translation tools could potentially map target language embeddings to an even better initialization for subsequent alignment with English, further reducing disparity.