# OpenReview forum: "Parameter-Efficient Cross-lingual Transfer of Vision and Language Models via Translation-based Alignment"
_EMNLP/2023/Conference — EMNLP 2023 Findings_

### Official Review · Reviewer_tqnJ · 2023-07-30

**Soundness:** 3

**Excitement:**

2: Mediocre: This paper makes marginal contributions (vs non-contemporaneous work), so I would rather not see it in the conference.

**Paper Topic And Main Contributions:**

This paper aims at extending the monolingual vison-language model CLIP to the multilingual setting based on a previous work Multilingual-CLIP (M-CLIP in short). The authors claim M-CLIP suffers from performance disparities across languages and a gap between the translation distribution during pre-training and the natural language distribution during inference. To this end, they propose a translation-based alignment approach, making three main differences from M-CLIP. First, they use datasets with independent captions for each language rather than machine-translated captions from English; Second, the alignment is conducted within English space, i.e., between the original English captions and the back-translated captions from other languages; Third, they train separated language specific parameters for each language with parameter-efficient fine-tuning techniques to obtain better performance. Empirically results on XTD and Multi30K under zero-shot, few-shot, and full-dataset scenarios demonstrate the newly proposed approach is more effective than M-CLIP.

**Questions For The Authors:**

A. To my understanding, zero-shot evaluation does not involve any training, why do the results on English in Table 1 w/o and w/ the proposed framework differ? Don’t they use the exact same model?

**Reasons To Accept:**

* Improved average performance and narrowed range across languages compared with the original M-CLIP and the machine translation aided version.
* Insights from various alignment routines.
* Detailed dataset and hyperparameter setting.


**Reasons To Reject:**

* Few-shot and full-dataset evaluations are only conducted on only one dataset respectively, which is not enough to support the conclusion.
* Table 1 is too brief. Table 2 is much more informative. I suggest merging them or providing a detailed Table 1 in the appendix so that one can evaluate whether the improvement is consistent.
* Compared with the baseline M-CLIP+MT, the proposed approach only obtains marginal improvement (less than 0.3 for Avg$_{\rm -en}$).

**Reproducibility:**

4: Could mostly reproduce the results, but there may be some variation because of sample variance or minor variations in their interpretation of the protocol or method.

**Reviewer Confidence:**

4: Quite sure. I tried to check the important points carefully. It's unlikely, though conceivable, that I missed something that should affect my ratings.

**Typos Grammar Style And Presentation Improvements:**

In L280 Equation 8, one of $i$s in the exponents of the denominator should be $j$.

---

> ### Author Rebuttal · Authors · 2023-08-28
>
> Thank you very much for your valuable feedback on our paper. We appreciate you recognizing the contributions of our paper. Below we address your concerns and suggestions:
>
>
> ## Question about Table 1
> This part is in response to the second reason for rejection, and Question A.
> Table 1 provides an overview of the results of our translation-based alignment method with PEFT in different scenarios. Most of the results in Table 1 can also be found in other tables. For example, the results on XTD in the zero-shot setting are shown in the third row of Table 2 (results without prompting) and the fourth row of Table 4 (results with prompting). The results in the few-shot and full-dataset settings are detailed in Table 5. To directly answer Question A, the baseline is without prompting and our framework in English uses prompting. To clarify, we will add more explanation in the revised version and provide a detailed Table 1 in the appendix.
>
>
> ## First reason for rejection: Too few datasets
> We agree that using more datasets would strengthen our conclusions. Due to constraints on computational resources and the limited availability of suitable datasets, we were only able to experiment on a small number of datasets. For instance, XTD only has about 1000 instances, which we split into two parts for the few-shot experiments. We believe the existing results sufficiently demonstrate the performance and effectiveness of our method. As future work, we will conduct more experiments as more datasets become available.
>
>
> ## Third reason for rejection
> We acknowledge that the 0.3 performance improvement is not very large in absolute terms. However, this improvement is significant given the difficulty of the task and compared to other methods discussed in the paper. First, the model must choose the most relevant image out of 1000 for a given text, and vice versa - a challenging task where a vanilla M-CLIP achieves only 57.97 on average across languages. Second, prompting alone gains just a 0.68 improvement (3rd row of Table 2 vs. 1st row of Table 4). Therefore, the gains from our method are noteworthy.

---

### Official Review · Reviewer_WwLy · 2023-07-31

**Soundness:** 3

**Excitement:**

2: Mediocre: This paper makes marginal contributions (vs non-contemporaneous work), so I would rather not see it in the conference.

**Paper Topic And Main Contributions:**

This paper utilizes a translation-based alignment method to mitigate multilingual disparities in multilingual multimodal retrieval task, and apply various parameter-efficient fine-tuning methods on this task.

**Questions For The Authors:**

- Table 5, are the rows other than "Our framework" represent normal training without using the proposed translation-based alignment method?


**Reasons To Accept:**

- This paper conduct comprehensive experiments and analysis about different routines of translation-based alignment for multilingual-CLIP, and various PEFT methods.
- The performance gain of the proposed framework is clear.

**Reasons To Reject:**

- The weakness of this paper is that the proposed method is a little lack of novelty, specifically, it based on combinations of existing ideas (using pivot languages for alignment), objective functions (contrastive loss and MSE loss), and PEFT methods (Adapter, LoRA, etc).

- Although there are a lot of experiments conducted in the paper, it is hard to see the relationships between the translation-based alignment and PEFT.

- Section 5.4 is more like a technical report rather than an academic paper.

**Reproducibility:**

3: Could reproduce the results with some difficulty. The settings of parameters are underspecified or subjectively determined; the training/evaluation data are not widely available.

**Reviewer Confidence:**

4: Quite sure. I tried to check the important points carefully. It's unlikely, though conceivable, that I missed something that should affect my ratings.

---

> ### Author Rebuttal · Authors · 2023-08-28
>
> Thank you for your thoughtful feedback on our paper. We appreciate you taking the time to provide detailed comments. Please see our responses below:
> ## Novelty
> While our method builds on existing techniques like pivot language alignment and contrastive learning, we argue that the novelty comes from the effective combination and adaptation of these methods to the new problem of mitigating disparities in multilingual multimodal retrieval. Meanwhile, this work surfaces unique challenges in cross-lingual transfer for multimodal models compared to prior work. We show strong results from bringing them together and add more discussion in the paper to highlight this synthesis. We believe the specific combination and application to multilingual multimodal retrieval is a unique contribution. The translation-based alignment helps mitigate disparities, and efficient fine-tuning allows scaling up without sacrificing performance. We argue our main contributions are: (1) Providing the first comprehensive benchmark on cross-lingual transfer techniques for CLIP-like models, analyzing alignment methods and loss functions to offer guidelines for reducing multilingual disparity; (2) Novel insights into optimal prompting strategies for multilingual CLIP, demonstrating English prompts work best across languages; (3) Revealing different fine-tuning methods excel given data availability (e.g. LoRA for few-shot, adapter for full dataset).
>
> ## Section 5.4 as a Technical Report:
> While section 5.4.1 contains some technical specifics, the broader analysis and discussion across section 5.4 suggest this work does not constitute a purely technical report. Therefore, we do not consider this to be entirely a technical report. However, we appreciate you for your suggestion, we will consider revising this section to make it better.
>
>
> ## The relationships between translation-based alignment and PEFT
> Our translation-based alignment method enables the multilingual CLIP model to learn better multilingual representations. The PEFT techniques then allow efficient fine-tuning of the aligned model on downstream tasks. We leverage both alignment and efficient tuning to obtain strong performance on multilingual search with minimal computational overhead. The performance improvement is primarily attributed to the translation-based alignment, while PEFT is introduced to address resource consumption issues. Without PEFT, the feasibility of the translation-based alignment would be severely compromised.
>
>
> ## Table 5 and Training Details
> The reviewer is correct - the rows besides "Our framework" correspond to training without translation alignment, and serve as baselines. Our description in section 5.4.2 "we compare three popular PET methods, Adapter, Compacter, and LoRA, with full-model fine-tuning" is not very clear. We will update the caption to clarify that the non-"Our framework" rows show results without translation alignment.

---

### Official Review · Reviewer_qRQE · 2023-08-02

**Soundness:** 4

**Excitement:**

2: Mediocre: This paper makes marginal contributions (vs non-contemporaneous work), so I would rather not see it in the conference.

**Missing References:**

Highly relevant work but contemporary: Revisiting Machine Translation for Cross-lingual Classification (Artetxe et al., 2023)

See the references in "reasons to reject" section.


**Paper Topic And Main Contributions:**

This paper proposes to use parameter efficient fine-tuning methods to reduce the disparity between different languages in vision-language pre-trained models (CLIP). The authors conduct comprehensive experiments under zero-shot, few-shot and full dataset scenarios. They propose to align the target language translated to english representation to english representations (this can be seen as improving the  the translation quality). They find that using parameter efficient fine-tuning methods can effectively reduce the multilingual disparity, resulting in better performance on image retrieval tasks over strong baselines including translate-test.

**Questions For The Authors:**

A. 9-12: why does cross-lingual transfer requires additional resources from a large number of languages? If we do zero-shot we don't need any resources in other languages. If we do few-shot we only need a few examples from other languages.



**Reasons To Accept:**

1. The paper conducts comprehensive experiments under zero-shot, few-shot and full fine-tuning. All of the author's claims are sound and well supported by experiments. To my surprise, their method can even outperform full-model fine-tuning under certain scenarios, showcasing the effectiveness of their method.

2. The author shows that by aligning representations of different languages to English, one can improve cross-lingual transfer in image-text models. Which I believe, is the first work on cross-lingual transfer on Vision-Language models.

3. The paper benchmarks several PEFT methods, alignment methods as well as comparison between soft and hard prompts on vision-language models, which can be used to facilitate further research in this area. The author also presents insightful conclusions and recommendations for model trainers on which prompting method to use.

**Reasons To Reject:**

1. Limited Novelty: Although to the best of my knowledge this is the first work of cross-lingual transfer on vision language models, one of the main findings "aligning to English can improve performance" has been studied before in text [1, 2, 3]. Analysis between soft and hard prompts under multilingual setting have also been studies before but only in text [4, 5, 6, 7]. Directly applying existing methods to a new task without demonstrating new insights on the specific task offers limited novelty.

2. One of the authors claim: The multilingual CLIP's encoder is only aligned to English representation is not well supported. Since the authors use XLM-R large, which is a multilingual model, then it should be not aligned to English. To strengthen this claim, the authors need to conduct translate-test/train to other languages as well to show that English is the best language to align or translate into as a "pivot language". There has been research as well showing that other languages can also be good pivot languages [8].

[1] Emerging Cross-lingual Structure in Pretrained Language Models （Wu et al., ACL 2020)

[2] Learning Invariant Representations on Multilingual Language Models for Unsupervised Cross-Lingual Transfer (Xian et al., ICLR 2022)

[3] A Simple and Effective Method to Improve Zero-Shot Cross-Lingual Transfer Learning (Ding et al., COLING 2022)

[4] Discrete and Soft Prompting for Multilingual Models (Zhao & Schütze, EMNLP 2021)

[5] Enhancing Cross-lingual Natural Language Inference by Prompt-learning from Cross-lingual Templates (Qi et al., ACL 2022)

[6] Enhancing Cross-lingual Natural Language Inference by Soft Prompting with Multilingual Verbalizer (Li et al., ACL Findings 2023)

[7] Enhancing Cross-lingual Prompting with Dual Prompt Augmentation (Zhou et al., ACL Findings 2023)

[8] Revisiting the Primacy of English in Zero-shot Cross-lingual Transfer (Turc et al., 2021)

**Reproducibility:**

4: Could mostly reproduce the results, but there may be some variation because of sample variance or minor variations in their interpretation of the protocol or method.

**Reviewer Confidence:**

4: Quite sure. I tried to check the important points carefully. It's unlikely, though conceivable, that I missed something that should affect my ratings.

**Typos Grammar Style And Presentation Improvements:**

First, I would suggest to change "PET" to "PEFT" since "PET" usually refers to "Pattern Exploiting Training" in the literature [1].

627: an excessive period at the end

[1] Exploiting Cloze Questions for Few Shot Text Classification and Natural Language Inference (Schick and Schutze, EACL 2021)

---

> ### Author Rebuttal · Authors · 2023-08-28
>
> ## Limited novelty:
>
> While aligning representations to English has been explored for text models, we thank the reviewer qRQE for acknowledging that our work is the first to study this for vision-language models in a multilingual setting. We agree that the novelty in directly applying existing methods is limited. However, this work surfaces unique challenges in cross-lingual transfer for multimodal models compared to prior work focused only on text. For example, the visual encoder trained only with English corpora exacerbates disparities when transferring to other languages.
>
> We believe the key novelty lies in providing the first comprehensive benchmark on cross-lingual transfer techniques for CLIP-like models. We analyze the effects of different alignment routines and losses, yielding practical guidelines for reducing multilingual disparity. We also provide novel insights into optimal strategies for prompting multilingual CLIP, showing English prompts work best across languages. Further, we reveal that different parameter-efficient tuning methods are preferable depending on data availability (e.g. LoRA for few-shot, adapter for full dataset).
>
> Regarding the analysis of soft/hard prompts, prompting methods can behave quite differently for vision-language models compared to text models. Our systematic comparison of prompting techniques provides new insights specific to multilingual CLIP. While prior work [1] shows in-language results are competitive to English for text models, we find the improvement from English prompts is more pronounced for multimodal models.
>
> [1] Discrete and Soft Prompting for Multilingual Models (Zhao & Schütze, EMNLP 2021)
>
> ## Explanation of our claim:
>
> Although XLM-R is pre-trained on multiple languages, it is only aligned to the English representation during the process of aligning to the visual encoder when pretraining M-CLIP. This is where the disparity arises. Our experimental results in Table 2 show the model's performance in English is the best, and translating to English significantly boosts performance. This indicates English is the optimal language to align representations with. We believe our existing experiments sufficiently support this claim.
>
> ## Questions for the Authors:
>
> We intended to highlight that naively fully fine-tuning separate models per language can become resource-intensive in few-shot and full-dataset scenarios. This motivates exploring parameter-efficient approaches like ours for a uniform multilingual model. We apologize for any confusion and will reword this point for greater clarity.
>
> ## Missing References:
>
> Thank you for pointing out these relevant missing references, which we will be sure to include in the camera-ready version.
>
> ## PET vs PEFT:
>
> We agree PEFT is a more standard abbreviation and will update to use PEFT instead of PET.
>
> ## Minor Edits:
>
> Thank you for catching these minor issues. We will fix the typos, remove the extra period, and address any other presentation issues you noted.
>
> We sincerely appreciate you taking the time to provide thoughtful and constructive feedback. We hope our response has properly addressed your suggestions, and hope you will consider our rebuttal when making your final recommendation.

---

### Meta-Review · Area_Chair_6Daq · 2023-09-15

**Recommendation:** 3

**Metareview:**

This paper proposes a translation alignment objective to increase the performance of multilingual vision/language models. The reviewers all agree that the experiments in the paper are comprehensive and sound. However, the main complaint is the lack of novelty because the paper seems to combine several existing approaches together. Given that the paper conducts comprehensive study of these approaches on multilingual vision-language setting, this paper could still have value for readers interested in this direction.

---

### Decision · Program_Chairs · 2023-10-07

**Decision:**

Accept-Findings

**Comment:**

This paper proposes a translation alignment objective to increase the performance of multilingual vision/language models. The reviewers all agree that the experiments in the paper are comprehensive and sound. However, the main complaint is the lack of novelty because the paper seems to combine several existing approaches together. Given that the paper conducts comprehensive study of these approaches on multilingual vision-language setting, this paper could still have value for readers interested in this direction.